# The Role of White Matter Dysfunction and Leukoencephalopathy/Leukodystrophy Genes in the Aetiology of Frontotemporal Dementias: Implications for Novel Approaches to Therapeutics

**DOI:** 10.3390/ijms22052541

**Published:** 2021-03-03

**Authors:** Hiu Chuen Lok, John B. Kwok

**Affiliations:** School of Medical Sciences, Faculty of Medicine and Health Sciences, University of Sydney, Camperdown, NSW 2006, Australia; hiu.lok@sydney.edu.au

**Keywords:** frontotemporal dementia, leukodystrophy, leukoencephalopathy, white matter, genetics, therapeutics

## Abstract

Frontotemporal dementia (FTD) is a common cause of presenile dementia and is characterized by behavioural and/or language changes and progressive cognitive deficits. Genetics is an important component in the aetiology of FTD, with positive family history of dementia reported for 40% of cases. This review synthesizes current knowledge of the known major FTD genes, including *C9orf72* (*chromosome 9 open reading frame 72*), *MAPT* (*microtubule-associated protein tau*) and *GRN* (*granulin*), and their impact on neuronal and glial pathology. Further, evidence for white matter dysfunction in the aetiology of FTD and the clinical, neuroimaging and genetic overlap between FTD and leukodystrophy/leukoencephalopathy are discussed. The review highlights the role of common variants and mutations in genes such as *CSF1R* (*colony-stimulating factor 1 receptor*), *CYP27A1* (*cytochrome P450 family 27 subfamily A member 1*), *TREM2* (*triggering receptor expressed on myeloid cells 2*) and *TMEM106B* (*transmembrane protein 106B*) that play an integral role in microglia and oligodendrocyte function. Finally, pharmacological and non-pharmacological approaches for enhancing remyelination are discussed in terms of future treatments of FTD.

## 1. Introduction

Frontotemporal dementia (FTD) syndromes are a heterogeneous group of proteinopathies, characterized by progressive degeneration of the frontal and/or temporal lobes. Clinically, they are divided into three subtypes. The behavioural variant FTD (bvFTD) is characterized by progressive deterioration of personality, social comportment and cognition [1], while the language variant progressive non-fluent aphasia (PNFA) is characterized by preserved comprehension but with poor expression of language. Semantic dementia (SD) is another language variant, with relatively preserved episodic memory and speech fluency, but lacking in content and with impaired comprehension of words [2]. Due to the heterogeneity of clinical presentations and underlying neuropathology, diagnosis and treatment of FTD has been challenging. Recently, our understanding of FTD has been greatly improved by the identification of causal genes that initiate and drive the neurodegenerative process. Previously, investigations of the pathogenic mechanisms of established causal FTD genes have focused on their impact on neuronal function. We now recognize that certain genes, such as granulin (*GRN*), have a significant impact on glia function, and mutation carriers show a specific pattern of white matter abnormalities. Further, genome-wide association studies (GWAS) and next-generation sequencing (NGS) of clinical and neuropathological FTD cohorts have identified a number of novel FTD genes associated with myelin or white matter abnormalities. In this review, we will synthesize current knowledge of the known FTD genes in terms of their impact on neuronal and glia neuropathology and function. We will examine the role of white matter in the aetiology of FTD and the clinical, neuroimaging and genetic overlap between FTD and conditions with white matter dysfunction (leukodystrophy and leukoencephalopathy). These insights could lead to a better understanding of pathogenic mechanisms and identify potential therapeutics for this complex disease.

## 2. The Genetics of Frontotemporal Dementia and Relation to Neuropathological Subtypes

Genetics is an important component in the aetiology of FTD, with approximately 10% of cases being autosomal dominant and 40% of patients having a family history of dementia [3]. BvFTD is highly heritable, with 37% of cases having strong family history of dementia (Goldman scores of ≤2), while the language variants PNFA and SD have lower incidence of positive family history (12.5% and 5.6%, respectively) [3]. Microtubule-associated protein tau (*MAPT*) [4], granulin (*GRN*) [5,6] and chromosome 9 open reading frame 72 (*C9orf72*) [7,8] are the three most common genes causal of FTD and together account for 30–50% of familial FTD [9,10]. In addition, tank-binding kinase 1 (*TBK1*) is found in 1–2% of total FTD cases [11,12]. Less common causal genes such as those for valosin-containing protein (*VCP*), chromatin-modifying protein 2B (*CHMP2B*), TAR-DNA-binding protein 43 (*TARDBP*), fused in sarcoma (*FUS*), coiled-coil-helix-coiled-coil-helix domain-containing 10 (*CHCHD10*) and triggering receptor expressed on myeloid cells 2 (*TREM2*) are known to contribute to <5% of all FTD cases [10,13,14]. Our group has recently identified *CYLD* as a causal gene for FTD and amyotrophic lateral sclerosis (ALS), but it appears to be a relatively rare cause of the disease [15,16].

The three FTD clinical sub-types display significant regional difference in grey matter structural atrophy and their connecting white matter tracts, with bvFTD showing specific prefrontal cortex and striatum grey matter atrophy and greater cerebellar white matter loss compared to other FTD subtypes [17]. Correspondingly, each causal gene is also associated with particular sub-types of FTD in terms of neuronal and glial pathology. The most common genetic causes of FTD will be discussed briefly in regard to their impact on neuropathology and their associated clinical and neuroimaging phenotypes.

### 2.1. Frontotemporal Lobar Degeneration-Tau (FTLD-Tau)

#### *MAPT* 

*MAPT* variants account for 20% of heritable FTD cases and are responsible for most familial cases with FTLD-Tau pathology [4], which is characterized by the presence of hyperphosphorylated tau inclusions, resulting from the disruption of tau binding to tubulin due to *MAPT* mutations [18,19]. Based on the biochemical composition and morphology of the inclusions, FTLD-Tau is further subtyped into Pick’s disease, corticobasal syndrome (CBS) and progressive supranuclear palsy (PSP), in which either three-repeat or four-repeat tau is primarily deposited in neurons [20]. Our group recently confirmed that in globular glial tauopathy (GGT), four-repeat tau is primarily deposited in oligodendrocytes and astrocytes [21].

The main clinical subtype for *MAPT* mutation carriers is the behavioral variant (bvFTD), which can be concomitant with a dominant Parkinsonism phenotype such as CBS or PSP [22,23,24]. Neuroimaging studies in these patients have shown relatively symmetrical anteromedial temporal lobe and orbito-frontal grey matter atrophy [25,26] that is responsible for the behavioral and semantic deficits in these patients.

### 2.2. Frontotemporal Lobar Degeneration-TDP (FTLD-TDP)

Approximately 50% of all FTLD cases are FTLD-TDP [27]. FTLD-TDP is characterized by the accumulation of tau-negative, ubiquitin-positive inclusions that are formed from ubiquitinated and hyperphosphorylated abnormal C-terminus fragments of TAR DNA-binding protein 43 (TDP-43) [28]. Based on the cortical distribution, intracellular location and morphology of these inclusions, FTLD-TDP neuropathology is further classified into types A to D [29,30]. To date, FTD genes associated with this form of FTD neuropathology include *GRN*, *C9orf72* and *TBK1*, whose features will be discussed below.

#### 2.2.1. *C9orf72*

As much as 85% of FTD cases caused by *C9orf72* expansions are associated with the bvFTD subtype [31], while 22.31% of *C9orf72* expansion carriers display additional degeneration of motor neurons or FTLD with amyotrophic lateral sclerosis (FTLD-ALS) [31,32]. The diffuse, widespread patterns of atrophy predominantly at the frontal and anterior temporal lobes, parietal lobes and the cerebellum lead to a broad range of clinical, cognitive and psychiatric symptoms [26,33]. Although the repeat expansion in *C9orf72* is the most common cause of FTD and ALS in Europe and North America [34], it is extremely rare in Asian and the Middle Eastern cohorts [34,35,36,37]. Our group has confirmed the relatively high prevalence of the *C9orf72* expansion in Australian and Spanish populations [38,39].

The neuropathology associated with *C9orf72* expansion is a combination of typical ALS and Type B or pure Type A [32,40,41,42] TDP neuropathology. *C9orf72* expansion was shown to impair the ability to degrade TDP-43, leading to TDP-43 accumulation in neurons and, occasionally, oligodendrocytes [43]. In addition, the glycine–arginine repeat protein [poly(GR)] translated from the expanded G4C2 repeats was shown to sequester full-length TDP-43, potentially resulting in the formation of TDP-43 inclusions [44]. Both RNA foci and dipeptide repeats predominantly impact on neuronal cell types [7].

#### 2.2.2. *TBK1*

Loss-of-function mutations in *TBK1* are found in ALS, FTD and ALS-FTD cases [12], with bvFTD being the most common observed clinical phenotype [12,32,45]. However, *TBK1* variants are also associated with PNFA [45] and CBS [46]. Phenotypic heterogeneity is also observed for this gene, with multiple clinical phenotypes being observed within the same family [32,46]. *TBK1* mutations are known to give rise to both FTLD-TDP subtype A [47] and B [12,32,45,48] neuropathology, with reports of severe neuronal loss [12,45,46], neuronal cytoplasmic inclusion [12,47] and gliosis [45,46].

#### 2.2.3. *GRN*

The majority of *GRN* mutations are loss-of-function [24] and account for 5–20% of familial FTD cases and 1–5% of apparently sporadic FTD [6], giving rise to Type A FTLD-TDP neuropathology [49]. This FTLD-TDP subtype, is primarily characterized by neuronal nuclear and/or cytoplasmic inclusions and dystrophic neurites [50], as well as oligodendrocyte inclusions [51]. To date, several mechanisms which link *GRN* mutations to FTLD-TDP pathology have been proposed: progranulin-mediated caspase 3-dependent cleavage of TDP-43 [52], sortilin-mediated progranulin endocytosis [53] and impaired autophagy [54] in the neurons. Of note, *GRN* is a growth factor and modulator of microglia, and mutation carriers have enhanced microglia activation [50] and gliosis [55,56] upon neuropathological examination.

FTD associated with *GRN* is highly heterogeneous in age of onset and clinical presentations, even for the same variant [57]. *GRN* variants have been associated with parkinsonism, CBS and Alzheimer’s disease (AD) [58,59]. Over 50% of *GRN* variant carriers present bvFTD followed by PNFA, and less than 10% are affected by CBS, AD and schizophrenia [24,60]. Indeed, multiple studies have shown that PNFA is more common in *GRN* variant carriers than in carriers of other mutations [24,31,32], consistent with the FTLD-TDP subtype A, which is usually associated with bvFTD and PNFA [61]. Neuroimaging of mutation carriers revealed asymmetrical atrophy in the frontal, temporal and interior parietal lobe [25,26,58]. Of interest, white matter hyperintensities and white matter lesions are often seen on MRI brain scans of *GRN*-associated FTD [62,63,64,65,66], and this aspect will be further discussed in later sections.

## 3. White Matter Changes in FTD

White matter involvement in FTD has been widely reported [26,66,67,68,69,70,71,72,73,74,75,76,77], suggesting a role for glial cells in the aetiology of the disease. White matter comprises the axons and their myelin sheath as well as the glia (oligodendrocytes, astrocytes and microglia) [78] and is crucial for cognitive and electrophysiological functions by coordinating rapid communication between different regions of the brain [79]. The myelin sheath is produced by oligodendrocytes wrapped around axons [80] (Figure 1), and disruption of this process leads to the absence of myelin production or its degeneration [79].

White matter alteration is prevalent in early symptomatic phase [74,75,81] and familial pre-symptomatic FTD [82,83,84,85]. White matter atrophy has also been reported in neuropathologically confirmed cases [86,87]. Diffusion tensor imaging has demonstrated selective reduction in white matter in the superior longitudinal fasciculus that interconnects the frontal and occipital and the temporal and parietal regions [74], which was found to correlate with behavior deficits in patient cohorts [74]. Similarly, Agosta et al. identified distinct structural network changes in white matter associated with specific neurobehavioral components, e.g., white matter changes in the frontotemporal regions are linked to apathy and impulsivity in bvFTD [88].

Consistent and widespread white matter tract pathology has been reported in bvFTD patients, compared to clinically normal subjects and those with AD and other forms of dementia [81,89]. A subgroup of the cohort carrying *MAPT* and *C9orf72* mutations exhibited a relatively discrete and distinctive white matter profile that showed alterations in white matter within the left anterior temporal pole when compared to control and Alzheimer’s disease subjects [81]. Further longitudinal studies of pre-symptomatic and symptomatic carriers of *C9orf72*, *MAPT* and *GRN* mutations have revealed early and widespread loss of white matter integrity in pre-symptomatic FTD, and there were clear genotypic “fingerprints” of white matter loss associated with each disease gene [82,83,84,85]. Since white matter is primarily composed of lipids, the imaging abnormalities observed in FTD brains could reflect the consequences of lipid metabolism dysregulation.

## 4. Lipid Metabolism in White Matter and Relevance to FTD

The human myelin lipidome consists of 700 different lipid moieties, 60% of which are species that are classified as phosphatidylcholines, phosphatidylethanolamines, sphingomyelins, cerebrosides and sulfatides [90]. The molar ratio of the major lipid components of myelin is kept constant, with cholesterol, glycerophospholipids and glycosphingolipids (in particular, galactocerebrosides) present at a ratio of around 4:4:2 [91], whose disruptions have frequently been associated with myelin dysfunction.

Lipid molecules in the brain are involved in energy storage, maintaining the structural integrity of the nervous system, signal transduction, modulation of membrane fluidity, trafficking of membrane proteins/receptors, cytoskeletal organization and neurotransmission [92,93]. The majority of brain cholesterol (~70%) is found in the myelin sheath; cholesterol is also an essential component of synapses and dendrites. Unsurprisingly, changes in brain lipid metabolism have been linked to AD [94,95,96,97], Parkinson’s disease (PD) [98,99], Huntington’s disease [100,101,102,103], ALS [104,105] and multiple sclerosis [106].

Changes in brain lipid metabolism are also evident in FTD. Lipidomic analysis performed on the plasma of bvFTD and AD patients has revealed a significant increase in triglycerides and decreases in phosphatidylserine and phosphatidylglycerol in bvFTD patients compared to AD patients [107,108]. Further, there were strong correlations between changes in certain lipids associated with mitochondrial energy production, proinflammatory pathways or oxidative damages and the pathophysiological changes associated with FTD, which can potentially be used as FTD biomarkers [109].

## 5. Granulin Mutations: A Model for Understanding the Role of Lipid Dysregulation and White Matter Changes in FTD

*GRN* variants frequently lead to extensive gliosis and loss of myelin in the underlying white matter [55,56,110,111]. White matter hyperintensities and white matter lesions are often seen on MRI scans of the brains of patients carrying *GRN* mutations [62,63,64,65,66]. Lipidomic analysis has shown dosage-dependent differences in brain lipids from humans and transgenic mice with progranulin deficiency [112]. These observations point to alterations in brain lipid metabolism as a result of *GRN* mutation. Indeed, homozygous carriers of *GRN* mutations develop young adult-onset neuronal ceroid lipofuscinosis (NCL), a lysosomal storage disease characterized by lipofuscin deposition [113,114].

While the exact mechanism that links *GRN* mutations to dysregulation of lipid metabolism remains to be elucidated, the presence of the lipofuscinosis phenotype in *GRN* mutation carriers implicates lysosome regulation [115,116]. Indeed, sortilin, the endocytotic receptor responsible for lysosomal progranulin trafficking [53], has been shown to be a regulator of lipoprotein metabolism [117]. In addition, the loss of granulin has been shown to impact lipid metabolism by altering saposins, cofactors for lysosomal lipid hydrolases, in the brain tissue of *GRN* mutation carriers and *Grn*^−/−^ mice [112]. It was also reported that progranulin facilitates the lysosomal trafficking of prosaposin, the precursor of saposin which is essential for the lysosomal degradation of glycosphingolipids [118]. Of interest, the activity of beta-glucocerebrosidase, a sphingolipid-metabolizing enzyme, was impaired in FTD phenotypic *Grn*^−/−^ mice and FTD patients carrying *GRN* mutations [119].

## 6. Leukoencephalopathies/Leukodystrophies as Part of the FTD Spectrum

Leukoencephalopathy is a collective term for a group of heterogeneous primary white matter disorders that can be either acquired or hereditary. Leukodystrophies—a term that derives from “leuko” = white, “dys” = lack of and “trophy” = growth—are heritable leukoencephalopathies. Leukodystrophy is defined as “heritable disorders affecting the white matter of the central nervous system, sharing glial cell or myelin sheath abnormalities, the neuropathology of which is primarily characterized by the involvement of oligodendrocytes, astrocytes and other non-neuronal cell types, although in many disorders the mechanism of disease remains unknown, and in other cases is suspected to include significant axonal pathology” [120].

Leukodystrophies are highly variable in age of onset and clinical manifestations [120,121], which include dementia, movement disorders, ataxia and upper motor neuron signs concomitant with hyperintense signal abnormalities in the brain/spinal cord on T2-weighted MRI [122]. Leukodystrophies can be broadly divided into hypomyelinating leukodystrophies (HLD) and demyelinating leukodystrophies, referring to defects in myelin developments and progressive deterioration of normally developed myelin, respectively [123].

The diagnosis of leukodystrophies is slow and challenging, and the prognosis of patients is poor as a result of very limited therapies, which are only beneficial in the early onset of the diseases [121]. Recent advances in sequencing technology have led to the rapid identification of the underlying gene defect and a more accurate diagnosis of leukodystrophies. Indeed, as a result of the emergence of whole-exome sequencing, the percentage of leukodystrophy cases without specific diagnosis dropped from about 50% in 2010 to 20–30% in 2016 [124,125]. The genetic diagnosis can be highly beneficial in terms of identifying therapeutic targets, thus allowing, for example, the administration of the relevant recombinant enzyme (enzyme replacement therapy), as done for the treatment of metachromatic leukodystrophy due to *Arylsulfatase A* (*ARSA*) mutations [126].

It is of interest to note that while the major FTD causal genes such as *MAPT* [4], *GRN* [5,6] and *C9orf72* [7,8] were identified using genetic linkage studies of large pedigrees, these approaches have limited capacity for the identification of variants with modest effect sizes [127]. These limitations have been bridged by the emergence of GWAS and NGS for the identification of novel disease genes and pathways associated with FTD. Indeed, GWAS and NGS performed on FTD patients have identified a significant portion of FTD risk genes involved in lipid metabolism and leukodystrophy/leukoencephalopathies, thus confirming a link between dysregulation in lipid metabolism, white matter dysfunction and FTD.

## 7. GWAS and Susceptibility Loci for FTD

GWAS is based on the rationale of the “common disease, common variant” hypothesis that attributes common diseases to allelic variants that are present in more than 1–5% of the population [127]. The association between multiple allele or genotype frequencies within the whole genome and pathological traits is rapidly determined using specific genotyping arrays [128]. To date, multiple GWAS performed on FTD cohorts [129,130,131,132,133,134,135,136,137,138,139,140,141,142] have identified genes with diverse biological functions associated with disease, as well as shared pathways with other neurodegenerative diseases such as AD and PD [136] (Table 1). A notable portion of these genes have links to lipid metabolism or glial function: mutation of *TMEM106B* causes childhood-onset leukodystrophy, a hereditary white matter disease [143], and *APOE*, *TOMM40* and *LRRK2* are involved in lipid metabolism [144,145,146,147].

## 8. NGS and Rare Variants in FTD

The “rare variants” hypothesis postulates that rare variants, primarily defined as having allele frequency <0.001 in the general population, could provide an explanation for a portion of missing heritability observed in twin studies [151] and impact on clinical phenotypes such as severity and age of onset [152]. NGS can rapidly screen the whole genome (whole-genome sequencing, WGS), specific loci or selected candidate genes (targeted sequencing), or exons of all coding genes (whole-exome sequencing, WES), thus enabling the parallel analysis of groups of interacting genes that contribute to the aetiology of a disease [151,153]. This approach has identified rare variants in known FTD genes as well as a number of rare variants in hitherto unsuspected genes with a role in lipid/white matter metabolism [37,45,151,152,154,155,156,157,158,159,160,161,162,163,164] (Table 2). For example, SNCA has been implicated as a lipid-binding protein [165,166,167], SORT1 is a known regulator of lipoprotein metabolism [117], ABCA7 is a lipid transporter [168], and the multi-faceted LRRK2 has roles in lipid storage and ceramide metabolism in the brain [144,145].

To date, there are over 60 genes with diverse biological functions that have been associated with leukodystrophy/genetic leukoencephalopathies [169,170], and there is growing evidence for a substantial overlap of leukodystrophy/leukoencephalopathy genes and FTD genes. Indeed, rare variants of *ARSA* [171,172], *CSF1R* [37,173,174], *TREM2* [156,163,175,176,177], *TYROBP* [178], *NOTCH3* [179] and *CYP27A1* [180] have been identified as potential genetic causes of patients with clinical features of FTD. A study by Sirkis et al. [181] showed that most genes implicated in both leukodystrophy and FTD risk are differentially expressed in FTLD postmortem brain and that leukodystrophy/FTD-associated genes are interconnected with genes regulating immunological function and lysosomal homeostasis [181]. Respectively, these findings highlight the involvement of white matter and the dysregulation of lipid metabolism in the pathogenesis of FTD. The most commonly reported leukodystrophy genes in FTD cases will be discussed in greater detail.

### 8.1. TMEM106B

TMEM106B belongs to the TMEM106 family of Type II transmembrane proteins, mainly localized in lysosomes [182,183,184]. TMEM106B variants are known risk modifiers for FTD [129,130,185,186], while a single TMEM106B mutation is responsible for five unrelated cases of hypomyelinating leukodystrophy [143,187]. To date, the exact mechanism through which TMEM106B variants are linked to hypomyelinating leukodystrophy and FTD risks remains to be fully elucidated, although studies have suggested that it involves TMEM106B-mediated lysosomal regulation/transport [188]. Of interest, TMEM106B-deficient mice showed impaired axonal lysosome transport and axonal autophagy [189] and myelin deficits [190], which suggest a role of TMEM106B in myelination.

### 8.2. TREM2 and TYROBP

In the brain, TREM2 is exclusively expressed in microglial cells, where it interacts with TYROBP to initiate signaling cascades that promote microglial cell activation, phagocytosis and microglial survival [202]. *TREM2* and *TYROBP* variants are found in patients with polycystic lipomembranous osteodysplasia with sclerosing leukoencephalopathy (PLOSL) or Nasu–Hakola disease (NHD), an autosomal recessive hereditary disease characterized by early-onset dementia and bone cysts [203,204], concomitant with atrophy of temporal white matter and loss of myelin and axon in the brain [205], although some NHD patients with FTD-like syndromes are reported to lack bone phenotypes [163,176,206]. Indeed, *TREM2* variants [156,163,175,176,177] and *TYROBP* variants [178] are associated with increased risk of disease in FTD patients.

Microglial cells are pivotal in remyelination by clearing myelin debris from sites of demyelination. Macrophages deficient in TREM2 were reported to have defects in the phagocytic pathways. A recent study of TREM2-deficient mice revealed cholesteryl ester accumulation in the brain and microglia, due to the transgenic mice’s inability to upregulate lipid metabolism genes. This led to an impaired ability of the microglia to phagocytose an influx of cholesterol brought about by chronic demyelination [207].

### 8.3. CSF1R

*CSF1R* encodes the colony-stimulating factor 1 receptor, a transmembrane tyrosine kinase receptor found in mononuclear phagocytotic cells and microglia in the brain that is crucial for microglial function [208,209]. *CSF1R* variants are associated with hereditary diffuse leukoencephalopathy (HDLS) with axonal spheroids and pigmented glia (ALSP) [193], bvFTD [37], CBS, AD, multiple sclerosis, leukoencephalopathy (CADASIL) and PD [193]. Heterozygous *CSF1R* variants were identified in patients with presentations similar to that of parkinsonism [210] and bvFTD [173,174]. Indeed, it has been postulated that 26.9% of histopathologically confirmed ALSP cases would have met the diagnostic criteria for “possible bvFTD”, while 11.5% could be diagnosed as “probable bvFTD” [211].

Neuropathologically, CSF1R-related leukoencephalopathy resulted in patchy white matter degeneration, featuring atrophy and confluent bilateral white matter hyperintensities, accompanied by loss of myelin and axons, presence of neuroaxonal spheroids and lipid-laden and pigmented macrophages [208,212,213] that were observed 6 years prior to symptoms onset [208].

### 8.4. CYP27A1

*CYP27A1* encodes sterol 27-hydroxylase, a mitochondrial cytochrome p450 oxidase involved in cholesterol metabolism and the synthesis of bile acids. Homozygous or compound heterozygous mutations in *CYP27A1* lead to cerebrotendinous xanthomatosis (CTX), a lipid storage disorder associated with a diverse range of neurological dysfunctions [214]. Of note, a homozygous mutation in *CYP27A1* was reported in a patient with clinical FTD, but without the typical CTX neuroimaging changes when examined using MRI [154]. Critically, the disease can be treated with bile salts prior to symptom onset [214], making it imperative to promptly perform a genetic diagnosis of mutation carriers and families.

## 9. Implications for Novel FTD Therapeutics

The consistent changes in white matter observed in FTD brains, as well as the genetic overlap between FTD and leukoencephalopathies/leukodystrophies, strongly support the hypothesis that therapies targeting white matter integrity, or remyelination, would be efficacious. Both processes of myelination and re-myelination after injury are mediated by oligodendrocytes [215]. The mature cells are derived from oligodendrocyte precursor cells (OPCs), which switch from a quiescent state to a regeneration phenotype in response to various biological signals. These include neuronal activity [216] and the expression of growth factors that drive OPC proliferation and promote survival [217]. This knowledge can be exploited to induce remyelination in the following scenarios.

### 9.1. Novel Targets for Gene-Specific or Pharmacological Intervention

The genetic overlap between leukoencephalopathies/leukodystrophies and FTD has opened up exciting opportunities for FTD therapeutics. *TMEM106B* is one of the genes that straddle the genetic spectrum of FTD and leukodystrophy, and knock-down of its expression appears to have a protective role in myelination [190]. Genetic ablation of *TMEM106B* expression was found to be effective in rescuing the FTD-like phenotypes in a transgenic mouse model carrying the *GRN* [218], but not the *C9orf72*, transgene [219]. However, this could reflect the differential white matter changes in *C9orf72* mutation carriers compared with *GRN* mutation carriers [82]. The concept of brain-region specificity can be exploited by pharmacological agents. BLZ945, an inhibitor of CSF1R kinase, was shown to enhance remyelination in the cortex/striatum [220], while PLX3397 increased myelination in the corpus callosum [221]. Recently, it was shown that TREM2 activation via agonistic antibodies accelerated microglial removal of myelin debris and enhanced remyelination [222]; also, prophylactic treatment with the compound was reported to prevent demyelination in mice [220].

### 9.2. Transcranial Magnetic Stimulation for Sporadic and Genetic Forms of FTD

Transcranial magnetic stimulation (TMS) is a non-invasive form of neural stimulation that applies local magnetic fields to generate electric currents in the brain, leading to an increase in neuronal activity [223] that is known to stimulate myelination [224]. TMS has consistently been shown to enhance myelination and secretion of neurotrophic factors in oligodendrocytes [225], to accelerate differentiation of an oligodendrocyte precursor cell line [226], to promote oligodendrocyte survival and differentiation in adult mouse brains [227] and to inhibit demyelination and increase neuronal and axonal survival [228].

Currently, clinical applications of TMS include diagnosis, monitoring and treatment of neurodegenerative diseases such as epilepsy [229], PD [230], AD [231] and ALS [232], as well as neuropsychiatric disorders including depression [233,234], schizophrenia [235,236] and bipolar disorder [237], without serious adverse effects [238,239]. Thus, the demonstrated effect of TMS on remyelination, together with its relative safety in clinical applications, makes TMS an ideal candidate for remyelination therapy in leukodystrophy/encephalopathy/FTD. Indeed, a randomized control trial of 70 FTD cases, of which 22 were *GRN*, *MAPT* and *C9orf72* mutation carriers, demonstrated efficacy, with significant increase of intracortical connectivity and improvement in clinical scores and behavioral disturbances in both symptomatic patients and pre-symptomatic mutation carriers [240].

## 10. Conclusions

Recent advances in sequencing technology have enabled the rapid discovery of novel FTD genes that enhance our understanding of this complex disease. While previous investigation into the pathogenesis of causal FTD genes has focused on their impact on neuronal functions, GWAS and NGS have revealed a number of novel FTD genes that affect both neurons and glia, whose mutations are associated with white matter abnormalities in carriers. In particular, our review highlights the extensive clinical and genetic overlap between FTD and leukoencephalopathy/leukodystrophy. This, in turn, has implications for pharmacological and non-pharmacological strategies to enhance oligodendrocyte numbers and the myelination process.

## Figures and Tables

**Figure 1 ijms-22-02541-f001:**
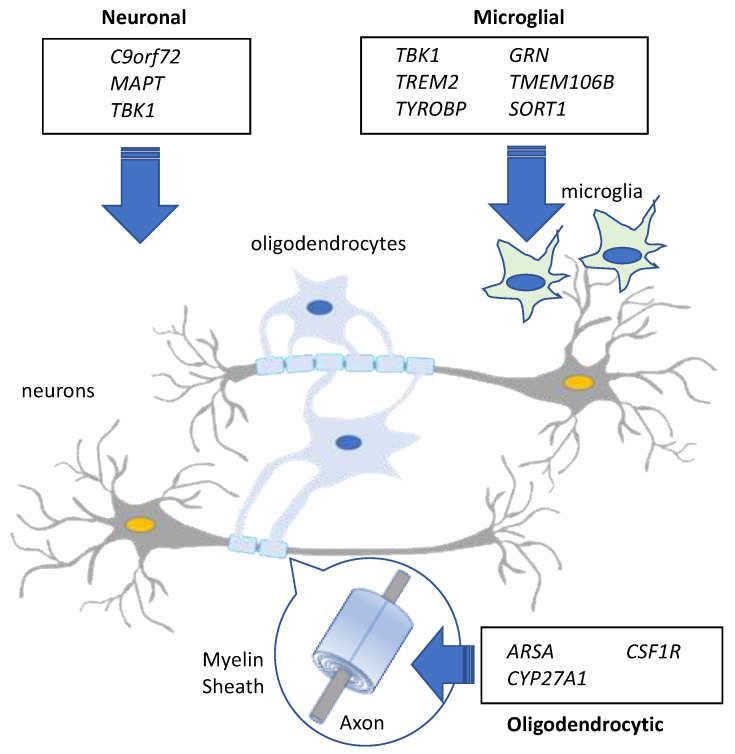
Schematic diagram of the relationship between neurons, oligodendrocytes and microglia. White matter comprises the myelin sheath covering the axons of neurons and is secreted by mature oligodendrocytes, as well as other glial cells such as microglia. Mutations and rare variants in established and candidate causal FTD genes can impact both neuronal and glial cells.

**Table 1 ijms-22-02541-t001:** Genes identified by genome-wide association studies (GWAS) that show association with FTD and impact on white matter.

Genes	Protein	Function(s)	White Matter Pathology/Disease
*TMEM106B* [129,131]	Transmembrane protein 106B	Unknown	Hypomyelinating leukodystrophy
*APOE* [136,141]	Apolipoprotein E	Lipid Metabolism	White matter hyperintensities upon MRI [148]
*LRRK2* [137]	Leucine-rich repeat kinase	Lipid Metabolism	Occasional LRRK2-immunopositive glia [149]
*RAB38* [135]	Ras-related protein Rab-38	Vesicle trafficking	Not described
*CTSC* [135]	Cathespin C	Activation of serine proteases in immune/inflammation	Not described
*TOMM40* [141]	Translocase of the outer mitochondrial membrane complex	Mitochondrial protein transport	Lower white matter integrity upon MRI [150]
*GFRA2* [132]	GDNF Family Receptor Alpha 2	Cell surface receptor for glial cell line-derived neurotrophic factor and neurturin	Not described

**Table 2 ijms-22-02541-t002:** FTD genes with rare variants identified by next-generation sequencing (NGS) and their role in white matter dysfunction.

Genes	Protein	Function	White Matter Pathology/Disease
*AARS2* [37]	Alanyl-tRNA synthetase 2	Translation	Leukodystrophy [191]
*ABCA7* [156]	Phospholipid-transporting ATPase ABCA7	Lipid transporter	Not described
*CCNF* [26]	Cyclin F	Cell cycle regulation	Not described
*CHCHD10* [192]	Coiled-coil-helix-coiled-coil helix domain containing 10	Mitochondrial function	Not described
*CSF1R* [193]	Colony-stimulating factor 1 receptor	Microglial function	Hereditary diffuse leukoencephalopathy [191]
*CTSF* [154]	Cathepsin F	Protein degradation	Leukoencephalopathy [191]
*CYLD* [194]	Ubiquitin carboxyl-terminal hydrolase CYLD	Autophagy, neuroinflammation	Widespread glia with CYLD-immunopositivity [15]
*CYP27A1* [154]	Cytochrome P450 family 27 subfamily A member 1	Cholesterol metabolism	Cerebrotendinous Xanthomatosis [191]
*LRRK2* [157]	Leucine-rich repeat kinase 2	Lipid metabolism	Occasional glia with LRRK2-immunopositivity [149]
*OPTN* [157]	Optineurin	Autophagy, membrane trafficking, cell cycle control, vesicle transport, NF-kB regulation	Not described
*PNF1* [157]	Profilin1	Regulation of actin polymerisation	Not described
*PSEN1* [154]	Presenilin 1	Proteolysis	Increased white matter hyperintensities [195]
*PSEN2* [154]	Presenilin 2	Proteolysis	Not described
*SNCA* [155]	Alpha-synuclein	Neuroprotection, neuronal differentiation, dopamine biosynthesis, maintenance of polysaturated fatty acids levels	Widespread oligodendrocytic inclusions in multiple system atrophy [196]
*SORL1* [156]	Sortilin-related receptor 1	Sorting and trafficking of intracellular proteins	Lower integrity of white matter tracts [197]
*SORT1* [161]	Sortilin 1	Protein trafficking; involved in glucose and lipid metabolism.	Not described
*SQSTM1* [164]	Sequestosome/p62	Autophagy	Widespread oligodendroglial pTDP-43 inclusions [198]
*TARDBP* [199]	TAR DNA-binding protein 43	Transcription and RNA splicing	-
*TREM2* [177]	Triggering receptor expressed on myeloid cells 2	Activation of macrophages, microglia and dendritic cell	Nasu–Hakola disease/polycystic lipomembranous osteodysplasia with sclerosing leukoencephalopathy [191]
*TYROBP* [178]	Transmembrane immune signaling adaptor	Macrophages and dendritic cells activation. Microglia activation in the brain	Nasu–Hakola disease [191]
*UBQLN2* [200]	Ubiquilin 2	Protein degradation, cell cycle regulation	Widespread demylination of white matter [201]

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
