# Peer review of "The Role of White Matter Dysfunction and Leukoencephalopathy/Leukodystrophy Genes in the Aetiology of Frontotemporal Dementias: Implications for Novel Approaches to Therapeutics"

_ijms, 2021, doi:10.3390/ijms22052541_

Round 1

Reviewer 1 Report

In this review the authors discussed and reported different issues linked to aetiology of frontotemporal dementias from white matter dysfunction to leukoencephalopathy/leukodystrophy, role of common variants and mutations in some genes (CSF1R, CYP27A1, TREM2, TMEM106) involved in microglia and oligodendrocyte function, and implications for novel approaches.

Although the importance of these topics is undisputed, I have some major points to address:

1) the authors should highlight better the added value of their work, underling the original points.

2) I suggest summarizing the first section (paragraph 2, line 49 page 2) describing just the main findings. I think that the Table 1 is superfluous.

3) On the other hand, the authors could extend the paragraph 3 concerning “white matter changes in FTD”, and find a clearer method to link the successive paragraphs.

4) The authors could modify the tables 2, 3 with more details in the description of functions of FTD risk genes underlying those more implicated in leukodystrophy/leukoencephalopathies and thus in white matter dysfunction. It is not pertinent to insert in the column “Other Associated Neurodegenerative Disease” AD or PD or ALS but rather reporting “leukodystrophy/leukoencephalopathies” as, for instance, the authors have done for “TMEM106B” and leukodystrophy.

5) Concerning the topic “Novel targets and intervention strategies”, “TMS” and “Multiple Sclerosis” are not connected with specific genes, thus these two topics seem to be detached whether the focus is the genetic implication.

6) Two much English mistakes. It is needed a strong revision of the English language.  

Author Response

We thank the reviewers for their insightful comments and have addressed the comments point by point in the following manner:

Reviewer #1:

In this review the authors discussed and reported different issues linked to aetiology of frontotemporal dementias from white matter dysfunction to leukoencephalopathy/leukodystrophy, role of common variants and mutations in some genes (CSF1R, CYP27A1, TREM2, TMEM106) involved in microglia and oligodendrocyte function, and implications for novel approaches.

Although the importance of these topics is undisputed, I have some major points to address:

  • the authors should highlight better the added value of their work, underling the original points.

  • We have published extensively on the impact of established and novel frontotemporal dementia genes, such as C9orf72, GRN, MAPT, and CYLD on neuropathology and clinical sequelae. In addition to references from our group that are already cited in the manuscript [reference # 4, 9, 15, 129, 132, 135, 136, 138, 158, 159], we have incorporated additional references from our group into the review article as follows:

  • Line 71: ‘Our group had recently identified CYLD as a causal gene for FTD and Amyotrophic lateral sclerosis (ALS), but it appears to be a relatively rare cause of the disease [15, 16].’

  • Line 89: ‘Our group recently confirmed that in globular glial tauopathy (GGT), 4-repeat tau is primarily deposited in oligodendrocytes and astrocytes [21].’

  • Line 133: ‘Our group have confirmed the relatively high prevalence of the C9orf72 in Australian and Spanish populations [38, 39].

Additional references:

  1. Oyston LJ et al. Reply: CYLD variants in frontotemporal dementia associated with severe memory impairment in a Portuguese cohort. Brain. 2020 Aug 1;143(8):e68. 

  1. Forrest SL et al. Are mutations in MAPT associated with GGT type III? Neuropathol Appl Neurobiol. 2020 Jun;46(4):406-409.

  1. Dobson-Stone C et al. C9ORF72 repeat expansion in clinical and neuropathologic frontotemporal dementia cohorts. Neurology. 2012 Sep 4;79(10):995-1001.

  1. Dobson-Stone C et al. C9ORF72 repeat expansion in Australian and Spanish frontotemporal dementia patients. PLoS One. 2013;8(2):e56899.

  • I suggest summarizing the first section (paragraph 2, line 49 page 2) describing just the main findings. I think that the Table 1 is superfluous.

  • We have deleted Table 1, and summarised the salient points in regards to heritability of the FTD subtypes in the text as follows:

  • Line 29: ‘Clinically, they are divided into three subtypes. Behavioural variant FTD (bvFTD) is characterized by progressive deterioration of personality, social comportment and cog-nition [1], while the language variant Progressive non-fluent aphasia (PNFA) is char-acterized by preserved comprehension but with poor expression of language. Semantic Dementia (SD) is another language variant and has relatively preserved episodic memory and speech fluency, but empty of content and with impaired comprehension of words [2].’

  • Line 60: ‘BvFTD is highly heritable with 37% of cases having strong family history of dementia (Goldman scores of = < 2), while the language variants PNFA and SD have lower inci-dence of positive family history of dementia (12.5% and 5.6% respectively) [3].’

  • On the other hand, the authors could extend the paragraph 3 concerning “white matter changes in FTD”, and find a clearer method to link the successive paragraphs.

  • We have clarified the concepts regarding white matter changes in the following manner:

  • Line 74: ‘The three FTD sub-types display significant regional difference in grey matter structural atrophy and their connecting white matter tracts, with bvFTD showing specific prefrontal cortex and striatum grey matter atrophy and greater cerebellar white matter loss com-pared to other FTD subtypes [17]. Correspondingly, each causal gene is also associated with particular sub-types of FTD in terms of neuronal and glial pathology. The most common genetic causes of FTD will be discussed briefly in regard to their impact on neuropathology, and their associated clinical and neuroimaging phenotypes.’

  • The authors could modify the tables 2, 3 with more details in the description of functions of FTD risk genes underlying those more implicated in leukodystrophy/leukoencephalopathies and thus in white matter dysfunction. It is not pertinent to insert in the column “Other Associated Neurodegenerative Disease” AD or PD or ALS but rather reporting “leukodystrophy/leukoencephalopathies” as, for instance, the authors have done for “TMEM106B” and leukodystrophy.

  • We have amended Table 2 and 3 (now Table 1 and 2) to include a specific section on white matter involvement in terms of report pathology and/or associated white matter disease.

  • Concerning the topic “Novel targets and intervention strategies”, “TMS” and “Multiple Sclerosis” are not connected with specific genes, thus these two topics seem to be detached whether the focus is the genetic implication.

  • We have extended the concept of the utility of TMS as a treatment sporadic and genetic forms of FTD as follows:

  • Line 442: ‘2 Transcranial Magnetic Stimulation for sporadic and genetic forms of FTD’.

  • Line 458: ‘. Indeed, a randomized control trial of 70 FTD cases, of which 22 were GRN, MAPT and C9orf72 mutation carriers, have demonstrated efficacy with significant increase of intra-cortical connectivity and improvement in clinical scores and behavioral disturbances in both symptomatic patients and pre-symptomatic mutation carriers [238].’.

  • Two much English mistakes. It is needed a strong revision of the English language.

  • The manuscript has now been edited by a native English speaker (Dr. Carol Dobson-Stone, University of Sydney, Australia) for clarity of language and grammatical/spelling errors. She has been acknowledged in the Authors contribution section as follows:

  • Line 989: ‘ We thank Dr. Carol Dobson-Stone (University of Sydney, Australia) for her help in the preparation of the manuscript.’

Reviewer 2 Report

The authors reviewed current knowledge of the known major Frontotemporal dementia (FTD)-related genes and potential novel genes associated with FTD, and discussed genetic overlap between FTD and leukodystrophy/leukoencephalopathy, and therapeutic approaches.

Overall the manuscript is well written. 

Minor comments:

-Abstract should be more concise.

-9.3 is out of focus. It would be better to focus on the FTD therapeutics.

Author Response

We thank the reviewers for their insightful comments and have addressed the comments point by point in the following manner:

Reviewer # 2:

The authors reviewed current knowledge of the known major Frontotemporal dementia (FTD)-related genes and potential novel genes associated with FTD, and discussed genetic overlap between FTD and leukodystrophy/leukoencephalopathy, and therapeutic approaches.

Overall the manuscript is well written. 

Minor comments:

  1. Abstract should be more concise.

  • We have re-edited the abstract to be more concise in the following manner:

Line 12: ‘Abstract: Frontotemporal dementia (FTD) is a common cause of presenile dementia and is characterized by behavioural and/or languages changes and progressive cognitive deficits. Genetics is an important component in the aetiology of FTD with positive family history of dementia reported in 40% of cases. This review synthesizes current knowledge of the known major FTD genes including C9orf72, MAPT and GRN, and their impact on neuronal and glial pathology. Further, evidence for white matter dysfunction in the aetiology of FTD, and the clinical, neuroimaging and genetic overlap between FTD and leukodystrophy/leukoencephalopathy are discussed.  The review highlights the role of common variants and mutations in genes such as CSF1R, CYP27A1, TREM2 and TMEM106B that play an integral role in microglia and oligodendrocyte function. Finally, pharmacological and non-pharmacological approaches for enhancing remyelination will be discussed in terms of future treatments of FTD.’

  1. 3 is out of focus. It would be better to focus on the FTD therapeutics.

-  We agree that the discussion on multiple sclerosis and associated therapeutics is too speculative at this stage, and have deleted the relevant paragraphs.

We hope that the amendments of the manuscript have addressed the concerns of the reviewers.

Yours sincerely,

Associate Professor John B Kwok

Round 2

Reviewer 1 Report

The authors replied to all my comments